# Development of an Optoelectronic Integrated Sensor for a MEMS Mirror-Based Active Structured Light System

**DOI:** 10.3390/mi14030561

**Published:** 2023-02-27

**Authors:** Xiang Cheng, Shun Xu, Yan Liu, Yingchao Cao, Huikai Xie, Jinhui Ye

**Affiliations:** 1School of Aerospace Engineering, Xiamen University, Xiamen 361005, China; 2Key Laboratory of Sensor Technology of Fujian Universities and Colleges, Xiamen 361005, China; 3Key Laboratory of Photoelectric Sensing Technology of Xiamen, Xiamen 361005, China; 4School of Ocean Information Engineering, Jimei University, Xiamen 361021, China; 5School of Integrated Circuits and Electronics, Beijing Institute of Technology, Beijing 100081, China; 6BIT Chongqing Institute of Microelectronics & Microsystems, Chongqing 401332, China

**Keywords:** MEMS mirror, optoelectronic integrated sensor high irradiance responsivity, high linearity, phase error

## Abstract

Micro-electro-mechanical system (MEMS) scanning micromirrors are playing an increasingly important role in active structured light systems. However, the initial phase error of the structured light generated by a scanning micromirror seriously affects the accuracy of the corresponding system. This paper reports an optoelectronic integrated sensor with high irradiance responsivity and high linearity that can be used to correct the phase error of the micromirror. The optoelectronic integrated sensor consists of a large-area photodetector (PD) and a receiving circuit, including a post amplifier, an operational amplifier, a bandgap reference, and a reference current circuit. The optoelectronic sensor chip is fabricated in a 180 nm CMOS process. Experimental results show that with a 5 V power supply, the optoelectronic sensor has an irradiance responsivity of 100 mV/(μW/cm^2^) and a −3 dB bandwidth of 2 kHz. The minimal detectable light power is about 19.4 nW, which satisfies the requirements of many active structured light systems. Through testing, the application of the chip effectively reduces the phase error of the micromirror to 2.5%.

## 1. Introduction

Over the past several years, structured light technology has seen great advancements and become one of the most promising three-dimensional (3D) shape measurement methods applied in diverse fields, including manufacturing, medical sciences, and entertainment [1,2,3,4,5]. Structured light systems can be classified into two categories: the passive type (e.g., stereo vision) and the active type. Active structured light systems are widely used because of the absence of limitations in light sources [6]. In particular, some structured light systems based on scanning MEMS micromirrors have shown distinctive potential for high-accuracy 3D shape measurement owing to their high speed, high precision, small size, and low cost. However, MEMS micromirrors usually generate small phase errors over time, mainly caused by the fatigue of the actuation arms of the micromirror as well as the environmental temperature, moisture and pressure. The phase errors would reduce the precision of an active structured light system considerably. Thus, phase detection is needed for MEMS micromirrors for structured light applications.

Several methods have been tried to monitor and compensate for the phase shifts of scanning micromirrors using piezoresistive [7,8,9,10,11] and capacitive sensors [12,13,14]. For example, C. Drabe et al. [8] and J. Grahmann et al. [9] both reported electrostatically driven piezoresistively sensed scanning micromirrors. S.Ju et al. demonstrated an electromagnetically driven piezoresistively sensed scanning micromirror [10]. Piezoelectric actuators are able to produce high force, but piezoelectric sensors are not suitable for phase monitoring and feedback control due to the lack of a dc response to reduce the steady-state error. C. Tsai et al. reported a biaxial electromagnetically driven piezoresistively sensed scanning micromirror fabricated in a CMOS process, but this method required considerable space in the MEMS micromirror system [11]. To realize a miniaturized monitoring scanning micromirror system, Hung et al. reported a capacitively transduced, closed-loop controlled scanning micromirror with shared comb electrodes for driving and sensing [15], but using shared electrodes for the slow-axis trajectory control produced instability due to the nonlinearity from electrostatic transduction [16]. M. Lee et al. examined a capacitive sensing strategy to quantify the shift between nominal and actual phase delays in the X and Y axes of a 2D electrostatic MEMS scanner [17]. The phase estimation showed an accuracy of 2.18°and 0.79°in the X and Y axes for motion sensing, respectively. However, capacitive sensing methods suffered from poor repeatability, low signal-to-noise ratios, and material limitations when using small MEMS devices in space-constrained applications. Various feedback control algorithms have also been proposed to reduce phase errors [18,19,20,21].

Optoelectronic integrated sensors have also been used to monitor scanning micromirrors, which can miniaturize the whole system in a simple way and have a large sensing range. Y. Liu et al. applied an optoelectronic integrated position sensor in the micromirror monitoring of Fourier-transform spectrometers, and the tilting angle was measured up to at least 5° with a resolution of 0.0067° [22]. Optoelectronic integrated sensors can be used to detect the phase of a scanning micromirror in an active structured light system, which is a good method to eliminate the phase error and improve the accuracy of scanning micromirror-based systems, but its irradiance responsivity is limited, and its linearity range is relatively small.

In this work, an optoelectronic integrated circuit (OEIC) chip based on a 180 nm CMOS process is developed and applied to detect a scanning MEMS micromirror’s phase. The new optoelectronic integrated sensor consists of a large-area photodetector (PD), which can increase the irradiance responsivity and improve the linearity range significantly. This paper is organized as follows. In Section 2, the MEMS micromirror, the optoelectronic integrated chip and the active structured light system are introduced. In Section 3, a model of the large-area PD is built in SILVACO and applied to simulate the optical spectrum response and AC characteristics of the PD. In Section 4, the receiving circuit design is described. In Section 5, the experimental setup and results are presented.

## 2. The Active Structured Light System and its Key Components

### 2.1. The Structured Light System Design

An active structured light system typically includes a control system, a projection system, and an image acquisition system [23]. The working diagram of the control system and the projection system is shown in Figure 1. The key device in this system is a MEMS scanning micromirror. A laser beam is shone on the micromirror. Combining the modulation of the laser with the angular scanning of the micromirror generates a fringe pattern that is projected on an object. The fringe pattern will be deformed by the morphology of the object. The deformed fringe pattern is captured by a camera, which is then processed to obtain depth information. In order to ensure the accuracy of the measurement results, the projection system needs to accurately project a set of sinusoidal grating fringes to the object of interest. Using a scanning MEMS micromirror to generate the grating fringes not only realizes accurate projection but also facilitates the miniaturization and integration of the measurement system.

A schematic of the projection system is shown in Figure 2, where a lens module converts a point laser into a line laser that is shone on a MEMS micromirror. The deflection of the mirror plate of the micromirror enables the line laser to scan in one dimension. The line laser is modulated to project sinusoidal grating fringes. The time when the micromirror reaches its maximum scan angle needs to be known in order to project sinusoidal grating fringes. In this paper, a thin glass is placed on the top of the micromirror to reflect the light beam back to an optoelectronic sensor chip, from which the maximum/minimum angle time can be calculated. The optoelectronic sensor chip is placed next to the MEMS micromirror chip, and both chips are bonded on the same solid slope, as shown in Figure 2. In this way, no bulky PCB-based amplification circuits for off-shelf PDs are needed, so this method can make the whole system much smaller.

### 2.2. The MEMS Micromirror

The MEMS micromirror is shown in Figure 3. The MEMS micromirror consists of two torsional beams, a 1.4 mm × 1.4 mm mirror plate, and four electrostatic comb drives. The device is fabricated on a 531 μm-thick silicon-on-insulator (SOI) wafer, which has a 1 μm-thick buried oxide layer, a 70 μm-thick device layer, and a 460 μm-thick handle layer. The first step of the fabrication process is to sputter a 200 nm-thick Au layer on the substrate surface. This Au layer is patterned to form the reflective mirror and bonding pads. Secondly, the torsion bars and comb drives are patterned and etched with a deep reactive ion etch (DRIE) process. This DRIE process is performed on the device layer from the front side of the SOI wafer and is stopped at the buried oxide layer. Then, the handle layer is etched by DRIE to form a back cavity. Finally, the buried oxide layer is removed by vapor HF to release the movable microstructures. More details about the design and fabrication of this MEMS mirror can be found in [24].

The resonant frequency of this MEMS micromirror is 1490 Hz. Upon operation, a 0–50 V square wave driving signal with a frequency slightly higher than the double resonant frequency can be applied between Pads #1 and #2. Pads #3~#6 are connected to the ground. For instance, the MEMS micromirror is driven by a square wave signal with 2983 Hz, which can guarantee a large field of view (FOV) of up to 80°.

### 2.3. The Optoelectronic Integrated Sensor

If the micromirror works exactly at the resonant frequency, the scanning can be easily disturbed by even small environmental variations. In order to obtain stable scanning, the micromirror generally works at a frequency slightly higher than the resonant frequency. Thus, the phase difference between the drive signal and the optical angular scan signal is not exactly 90°. A detection method is needed to calculate the phase difference and compensate it to ensure that the time point at which the laser starts to emit light matches the time point at the maximum/minimum scan angle of the micromirror. In this paper, an optoelectronic integrated sensor is placed next to the micromirror. Compared to those methods employing off-shelf PDs, this optoelectronic integrated sensor has higher SNR and a larger sensing range and does not need bulky PCB-based amplification circuits due to the integrated preamplifier.

The overall structure of the optoelectronic chip is divided into two areas: the photodetector (PD) and the receiving circuit. To reduce the large dark current noise brought by the large active area, a dummy diode with the same design is placed near the PD. The surface of the dummy diode is covered with metal to be insensitive to light. The receiving circuit handles the weak photocurrent signal and outputs a voltage signal.

### 2.4. The Control System and the Elimination of Phase Error

A field programmable gate array (FPGA) is used to control both the modulation of the laser (850 nm) and the scanning of the micromirror. Driven by a square wave signal with a frequency near the double resonant frequency, the optical scan angle of the micromirror presents a sinusoidal form, as shown in Figure 4. Ideally, the difference between the time the micromirror reaches its maximum scan angle and the rising edge of the driving square wave is fixed and known, so a fixed phase difference can be implemented in the FPGA-based control and data acquisition system. Unfortunately, the resonant frequency of a micromirror is affected by the environmental conditions as well as the fatigue of the actuation arms of the micromirror, causing a varying phase difference, as shown in Figure 4.

When the micromirror reflects the light beam to scan, the thin glass reflects the light beam to the optoelectronic integrated sensor next to the micromirror. As shown in Figure 2, the reflected optical beam 1 and beam 2 are scanned simultaneously with the movement of the micromirror. Optical beam 2 passes the OEIC chip four times in each scan cycle. When optical beam 2 passes the OEIC chip for the first time, the OEIC chip receives an optical signal, sends the corresponding electrical signal to the FPGA, and records the time as t_1_. When optical beam 2 passes the OEIC chip for the second time, the time is recorded as t_2_. As illustrated in Figure 4, there is a phase lag between the driving square wave and the optical scan angle, i.e., the micromirror does not reach the maximum rotation angle at the rising edge of the driving square wave. The time corresponding to this phase lag, ∆t, is the difference between t_2_ and half of the time during which the OEIC chip receives two consecutive optical signals. Assuming the period of the square wave signal is T, the phase lag time is given by
(1)$$∆t=t_{2}-\frac{t_{2}-t_{1}}{2}\pm \mathrm{nT}\;$$
where n is an integer. After delaying the rising edge of the driving square wave for a time of ∆t, the control system begins to project fringes. If there is any phase fluctuation from the micromirror scanning, ∆t will track that and thus can be compensated in real time.

## 3. Design of a Large-Area Photodetector in Sensor Chip

To obtain the time when the micromirror reaches its maximum scan angle, an optoelectronic sensor is placed next to the micromirror. Because most of the reflected-by-micromirror light beam shines through the thin glass to the measured object, the actual light power received by the photodetector is about less than 1 μW. Since the mV-voltage signal needs to be measured, the irradiance responsivity of the optoelectronic sensor must be designed to be over 80 mV/(μW/cm^2^), and the minimal optical signal of 20 nW must be picked up. The photosensitive area is designed to be 900 μm × 900 μm, which appropriately matches the reflected light spot with a diameter of 1 mm. The bandwidth of the chip should be higher than the resonant frequency of this MEMS mirror in the projection system. Based on our previous work on PDs integrated with standard CMOS technology [25,26], an N+/N-Well/P-sub structure is designed for the photodetector. The dark current of the effective PD can be partially balanced by a dummy PD that is covered by metal. The total current generated by the effective PD and the dark current generated by the dummy diode are subtracted in the receiving circuit, and the resultant current is the photocurrent.

As is shown in Figure 5, the N+/N-Well/P-sub structure of the Si photodiode is modeled by using the software SILVACO. The color represents the doping concentration. Due to the limitations of the grid definition in the simulation, the size of the model is scaled down. The model takes P-sub as the substrate and N+/Nwell as the central active area. The contact area between the P-sub and N-well forms a PN junction for simulation. The simulated optical spectrum response and AC characteristics of the PD are shown in Figure 6 and Figure 7, respectively, indicating that the bandwidth is as large as 2 kHz and the peak optical spectral response of the PD is at the wavelength of 850 nm, corresponding to a maximum responsivity of 0.3 A/W under an optical power of 0.5 µW/cm^2^ on the sensitive surface of the PD.

## 4. Realization of the Receiving Circuit in Sensor Chip

### 4.1. Overall Design of the Receiving Circuit

As shown in Figure 8, the receiving circuit is composed of two sampling resistors, two post amplifiers, an operational amplifier, and a reference current circuit. Both the PD and the dummy diode are connected to the receiving circuit. The resistance of the sampling resistor is set at the MΩ level, and the dark current of the PD is at the pA level. The current of the PD is converted into a voltage signal by a high-resistance sampling resistor, which is the input of one of the post amplifiers. The two post amplifiers have the same design with a unity gain negative feedback. The outputs of the two post amplifiers, V_tia_ and V_ref_, are the inputs of the differential operational amplifier, eliminating the dark current, as shown in Equation (2). When the photocurrent of the PD is in the range of 0~400 nA, the dynamic output voltage swing can reach 5 V, and the linearity is good.
(2)$$V_{\mathrm{out}}=\frac{R_{3}}{R_{4}}\left(V_{\mathrm{ref}}-V_{\mathrm{tia}}\right)=\frac{R_{3}}{R_{4}}\left(I_{\mathrm{op}}\times R_{1}-I_{\mathrm{dark}}\times R_{2}\right)\;$$
where I_op_ and I_dark_ are the photocurrent and dark current, respectively.

### 4.2. Design and Improvement of the Post Amplifier

The performance of the post amplifier determines the overall performance of the optoelectronic sensor. The dynamic output voltage swing and the open-loop gain of the post amplifier must be high. As shown in Figure 9, the input transistors Mp2, Mp3, Mn2, and Mn3 are in a complementary input differential pairs configuration, and the overdrive voltage of the output transistor is small, which is the key to obtaining the rail-to-rail output. Then, a folded-cascade amplifier [27] is used as the first stage of the post amplifier to achieve high gain. The gain of the post amplifier is calculated by Equation (3). It can be seen that the gain *A_v_* is approximately proportional to the output impedance of Mp10 and Mn10.
(3)$$A_{V}=g_{mp1}\left[g_{mp7}r_{op7}r_{op5}//g_{mn7}r_{on7}\left(r_{op3}//r_{on5}\right)\right]\times \left(g_{mn10}+g_{mp10}\right)\left(r_{op10}//r_{on10}\right)\;$$
where *g_mx_* is the transconductance of the corresponding MOS transistor, *r_ox_* is the equivalent resistance of the corresponding MOS transistor under the channel length modulation effect, and λ is the channel length modulation parameter.

The AC simulation of the post amplifier is shown in Figure 10. The open-loop gain of the post amplifier is 117.2 dB with a power supply of 5 V. Its bandwidth is about 5.7 MHz, and its phase margin is 66.5°, which satisfies the need for high responsivity and linearity. In addition, the overall noise of the circuit is $6.4\;\mathrm{nV}/\sqrt{\mathrm{Hz}}$, and the sample resistors R_1_ and R_2_ contribute nearly half of this noise. This noise can be reduced by a parallel-connected capacitor or using a sample-hold circuit.

However, when both the effective PD and the dummy diode output a dark current of pA level and the resistance value of the sampling resistor is at the MΩ level, the output of the chip is at an abnormal potential. In this paper, the above problem is avoided by changing the output offset of the post amplifier connected with the dummy diode. The schematic design is shown in Figure 11. Usually, the amplifier has the same number of input pairs. By adjusting the circuit, the number of Mn1 is *n* − *m*, and the number of Mn2 is *n*. The currents of Mn1 and Mn2 tubes are given in Equations (4) and (5), respectively. When *I_n_*_2_ is greater than *I_n_*_1_, the value of *I_n_*_2_ will eventually approach the value of *I_n_*_1_ due to the current mirror structure, which causes V_out_ to drop. This method effectively solves the problem of abnormal output caused by the same voltage at the positive and negative input terminals of the operational amplifier.
(4)$$I_{n1}=\frac{1}{2}\mu_{n}C_{ox}{\left(V_{gs1}-V_{th1}\right)}^{2}\left(1+\lambda V_{ds1}\right)\left(n-m\right)\;$$
(5)$$I_{n2}=\frac{1}{2}\mu_{n}C_{ox}{\left(V_{gs2}-V_{th2}\right)}^{2}\left(1+\lambda V_{ds2}\right)n\;$$
where *V_gs_* is the gate-to-source voltage, *V_th_* is the threshold voltage, and *V_ds_* is the gate-to-drain voltage.

### 4.3. Reference Current Circuit

In this work, the reference current circuit is used to provide a stable reference current, *I_bp_*, for the amplifiers. The reference current circuit diagram is shown in Figure 12. The bandgap circuit provides a stable reference voltage, *V_bg_*. The transistors M1 and M2 work as a current mirror, the OP works as the error comparator, and a feedback resistance *R*_1_ is introduced to form a negative feedback, which improves the stability of the reference current circuit when the supply voltage is not stable. The PSR simulation diagram of the reference current circuit is shown in Figure 13. The PSR of the reference current circuit reaches 85.4 dB at 2 kHz. The reference current can be calculated by Equation (6).
(6)$$I_{bp}=\frac{V_{bg}}{R_{1}}\;$$

It can be seen that the reference current is approximately proportional to *V_bg_* and 1/*R*_1_. Here. *V_bg_* is set at 1.2 V. *R*_1_ can be adjusted to obtain the proper *I_bp_*.

### 4.4. Improvement of the Output Offset

The offset is common in circuit design, including systematic offset and random offset. A post amplifier circuit is designed to avoid abnormal output, which results in a larger output offset on the chip, which results in a lower chip output swing. Figure 14 shows the structure of an improved operational amplifier with an offset resistor, *R*_0_. The output voltage is given in Equation (7). The output offset of the chip can be reduced by changing the value of *R*_0_.
(7)$$V_{out}=\left(V_{1}-V_{2}\right)\frac{R_{1}}{R_{2}}-\frac{V_{2}R_{1}R_{0}}{\left(R_{1}+R_{2}\right)R_{2}}\;$$

## 5. Results and Discussion

### 5.1. Experimental Platform of the Chip

The optoelectronic integrated sensor was monolithically implemented in a 180 nm CMOS technology. The size of the chip was 1.2 mm × 1.0 mm, and the size of the PD was 900 μm × 900 μm. The micrograph of the monolithic optoelectronic integrated chip is shown in Figure 15. The responsivity of the PD is very important for the design of the optoelectronic integrated chip. The measured spectral response curve of the PD is plotted in Figure 16. When the wavelength of the incident light is 850 nm, the responsivity of the PD is 0.32 A/W, and the measured dark current is 16 pA. The PD test results are close to the simulation results shown in Figure 7. Thus, the simulation model shown in Figure 5 is reliable and can be used for OEIC collaborative design.

Further measurement results show that the output voltage of the chip is nearly 5 V under an irradiance of 50 μW/cm^2^ of an 850 nm LED light source, and the irradiance responsivity of the chip is 100 mV/(μW/cm^2^). The curve of the output voltage versus irradiance is shown in Figure 17. The output voltage varies linearly with the irradiance from 0 to 50 μW/cm^2^.

### 5.2. Experimental Platform of the System

The projection system with a micromirror and a monolithic optoelectronic integrated sensor chip is shown in Figure 18.

Before the phase detection of the MEMS micromirror is realized, the optical power of the laser is measured and adjusted to the linear range that can be detected by the monolithic optoelectronic integrated chip. The maximum laser power is about 500 μW. Due to the high transmittance and low reflectivity of the thin glass, the measured power of the reflected light beam is less than 500 nW (as shown in Figure 2). The experimental results show that the OEIC chip can detect the minimum optical power of 19.4 nW. The output voltage reaches nearly 5 V when the optical power is 420 nW. The bandwidth of 2 kHz can meet the requirements of many active structured light systems.

The system sends and receives signals from the FPGA. A typical signal captured by the FPGA logic analyzer is shown in Figure 19. When the FPGA sends a modulation signal to the 850 nm laser, the sensor chip receives an optical signal for the first time and converts it as an electrical signal that is sent back to the FPGA, and the time t1 is recorded. When the sensor chip receives the optical signal for the second time, the time t2 is recorded. The times t3 and t4 are recorded when the falling edge of the square wave comes for the first and second time, respectively. Through the calculation method mentioned above (as shown in Equation (1)), the rising edge and falling edge of the electrical signal from the sensor chip are used to determine the phase difference time, ∆t. As shown in Figure 19, ∆t was 194.48 μs according to the calculation of the rising edge of the output pulse from the sensor chip, while ∆t was 207.57 μs according to the calculation of the falling edge.

There is no upper limit to how much the phase lag can be compensated as long as it is within one full period of the driving signal. In order to improve the robustness of the system, a number of time measurements are averaged in the FPGA to minimize the errors caused by the pulse edge jittering.

The laser was modulated, and a binary fringe pattern was projected on graph paper. The fringe pattern was manually adjusted by sending a phase delay ∆t through the serial port. Then, the pattern was observed and captured using an industrial camera. The optimal ∆t was found to be 199 μs. Experiments show that the results calculated from the rising edge are closer to the manual calibration value, with an error of 2.5%.

In the experiment, a binary fringe, coded as 10101010, was projected. The frequency of the driving square wave frequency was 2983 Hz. Without compensating for the phase lag, the fringe pattern cannot be projected correspondingly, and what the camera observes is haphazard, as shown in Figure 20a. After compensating for the phase lag based on the time difference extracted from the optoelectronic integrated sensor, the correct and uniform fringe pattern was obtained, as shown in Figure 20b. Note that environmental factors will also affect the performance of the proposed circuits but at a much smaller severity. Under the environmental conditions of this experiment, the proposed circuits are stable. For a large environmental change, this effect must be considered carefully.

## 6. Conclusions

A new optoelectronic integrated chip with a large-area PD for 850 nm structured light systems based on a MEMS micromirror is designed. The structure models of the PD are built in SILVACO. Based on the analysis of the characteristics of the PD, a monolithic chip integrated with the PD, whose area is 900 μm × 900 μm, is implemented in a 180 nm CMOS technology. The simulation results show that the gain of the amplifier on the chip is 117.2 dB, and the 3 dB bandwidth is 2 kHz. It is shown by the measured results that the responsivity of the PD is 0.32 A/W when the wavelength of the incident light is 850 nm. With a 5 V supply, the optoelectronic integrated chip has an irradiance responsivity of 100 mV/(μW/cm^2^), and its output voltage changes linearly with the irradiance, which satisfies the requirement of active structured light systems. In the test of the projection system in active structured light systems, by compensating for the phase lag based on the time difference extracted from the optoelectronic integrated sensor, the correct and uniform projected fringe pattern is obtained. The application of the chip effectively reduces the phase error to 2.5%. Laser projection systems incorporated with the chip can be applied to 3D cameras and machine vision.

## Figures and Tables

**Figure 1 micromachines-14-00561-f001:**
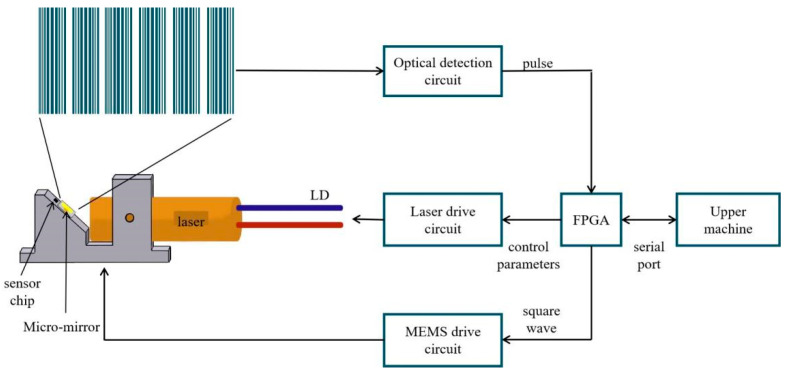
The architecture of the structure light system.

**Figure 2 micromachines-14-00561-f002:**
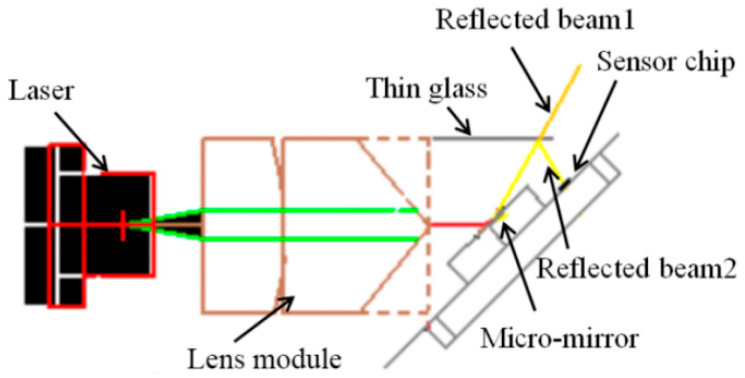
The structure of the micromirror-based projection system.

**Figure 3 micromachines-14-00561-f003:**
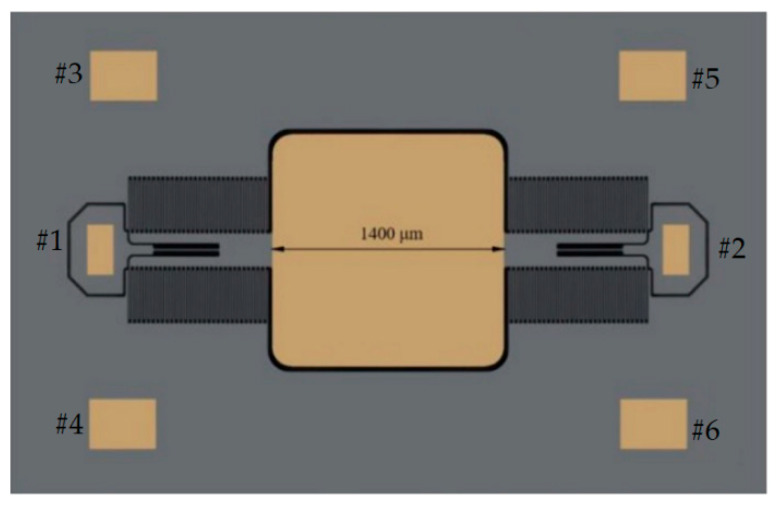
A top-view photo of the MEMS micromirror.

**Figure 4 micromachines-14-00561-f004:**
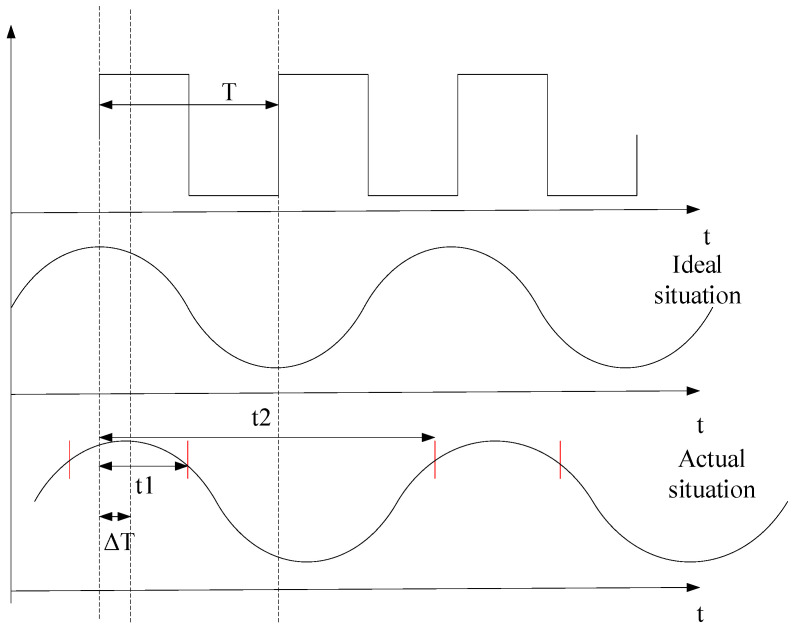
The phase diagram of the micromirror.

**Figure 5 micromachines-14-00561-f005:**
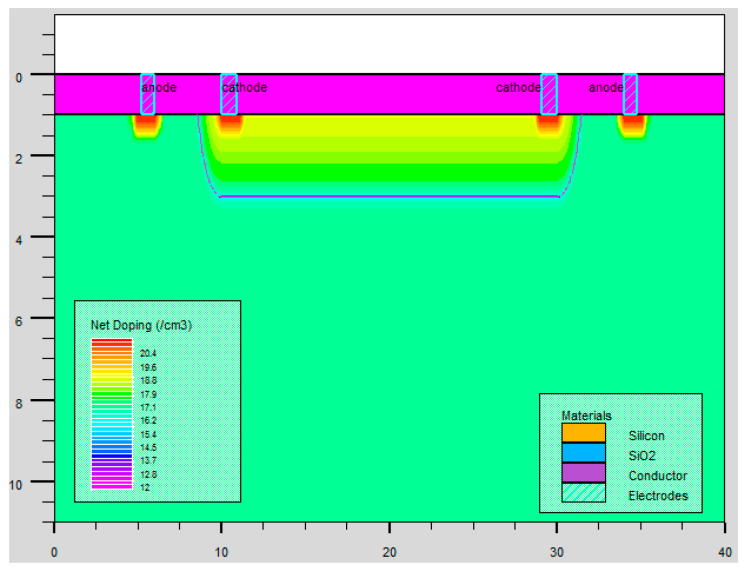
Two-dimensional structure of the photodiode.

**Figure 6 micromachines-14-00561-f006:**
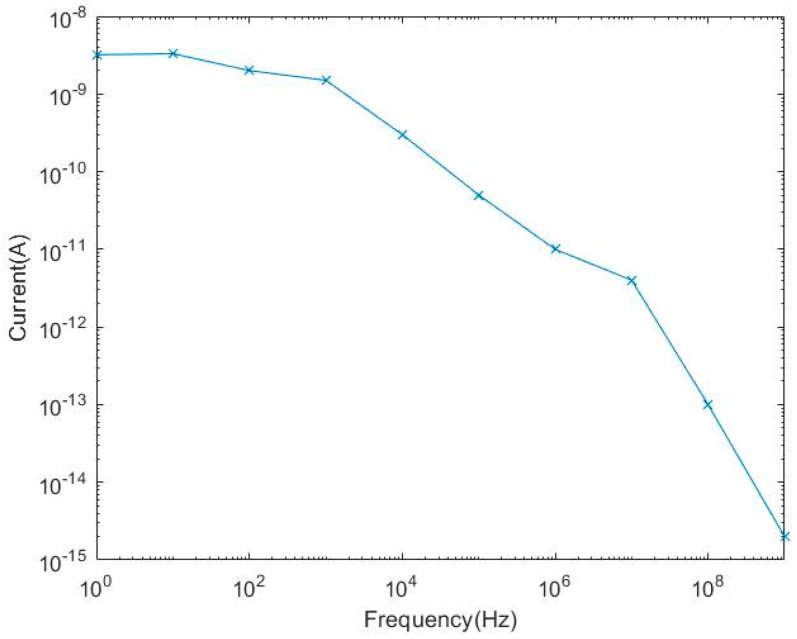
Simulated AC characteristic curve of the PD.

**Figure 7 micromachines-14-00561-f007:**
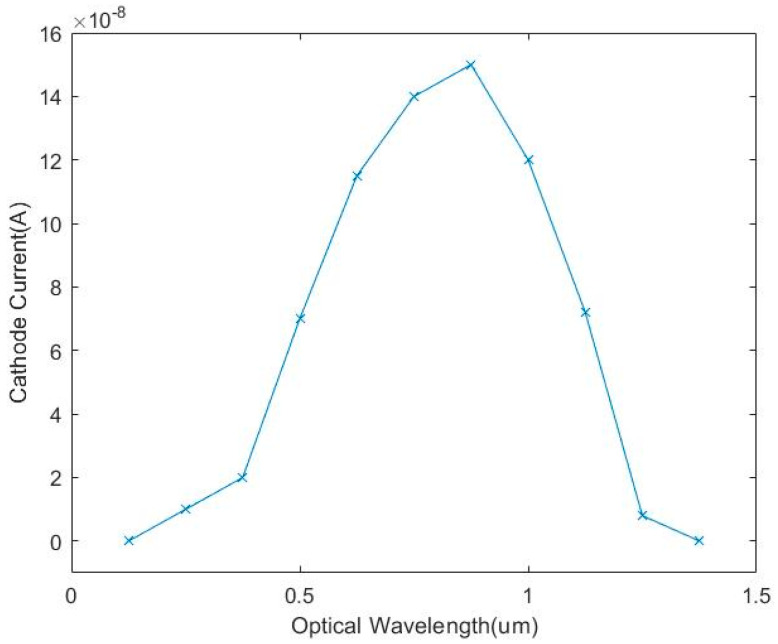
Simulated spectral response curve of the PD.

**Figure 8 micromachines-14-00561-f008:**
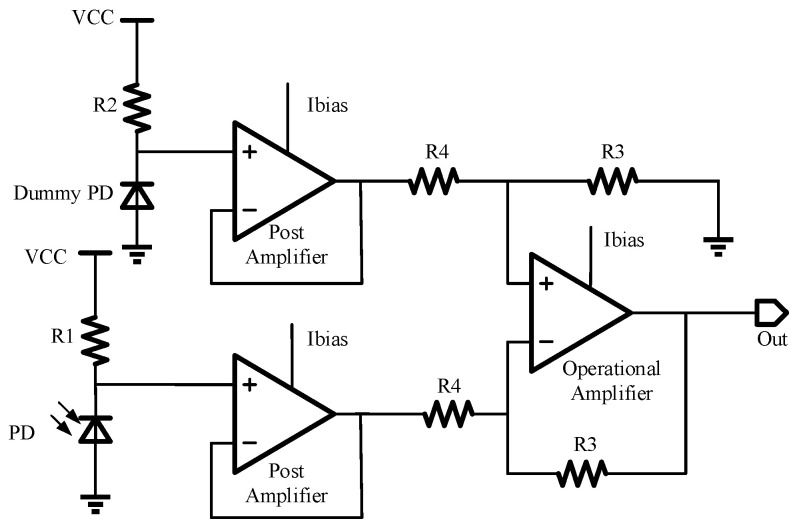
Schematic diagram of the receiving circuit.

**Figure 9 micromachines-14-00561-f009:**
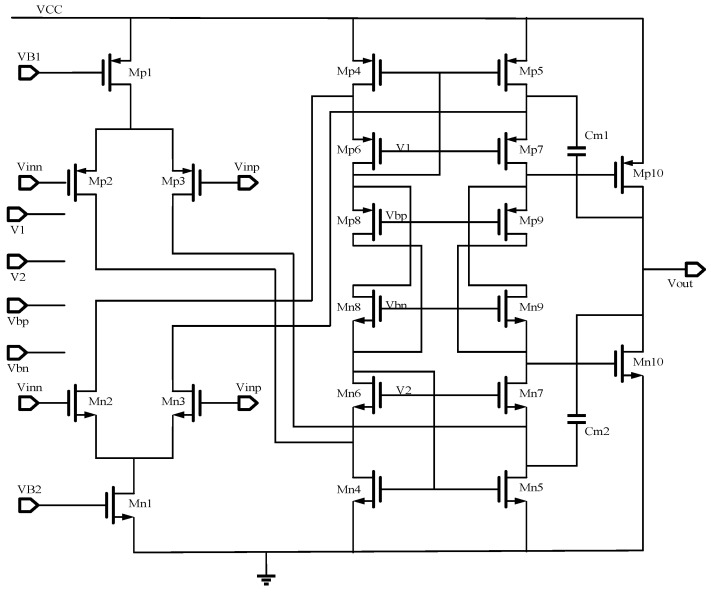
The topology of the post amplifier.

**Figure 10 micromachines-14-00561-f010:**
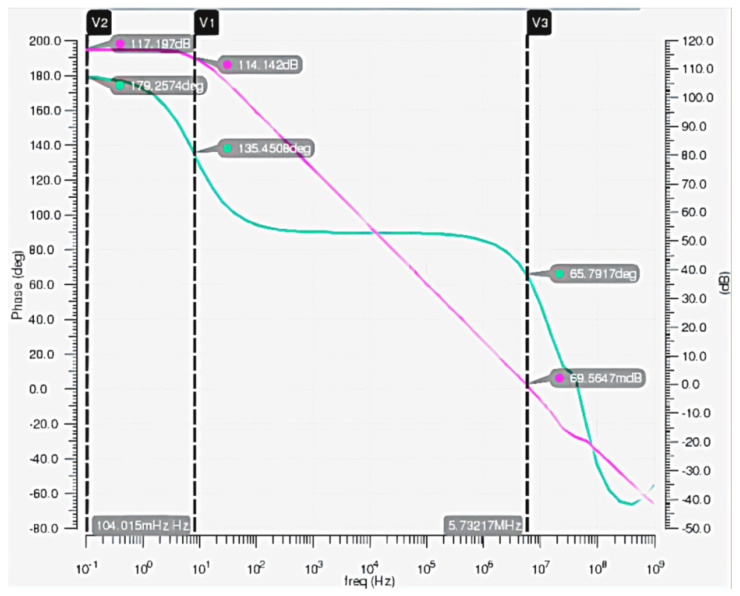
The AC simulation of the post amplifier.

**Figure 11 micromachines-14-00561-f011:**
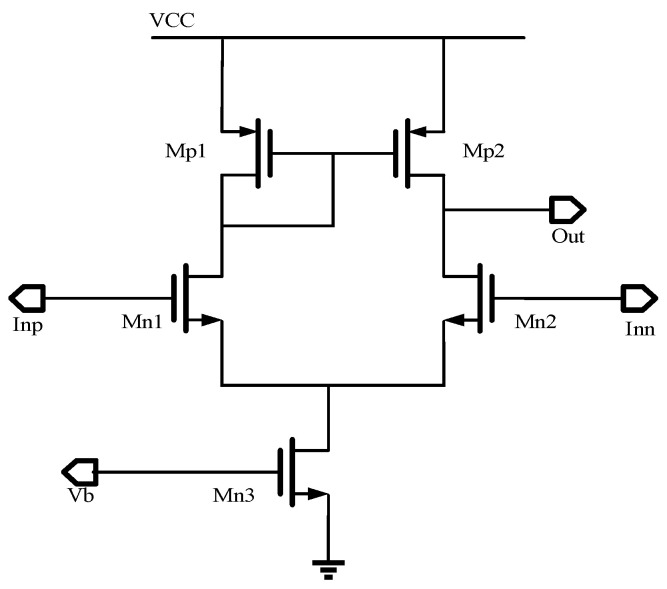
The principle of the improved amplifier.

**Figure 12 micromachines-14-00561-f012:**
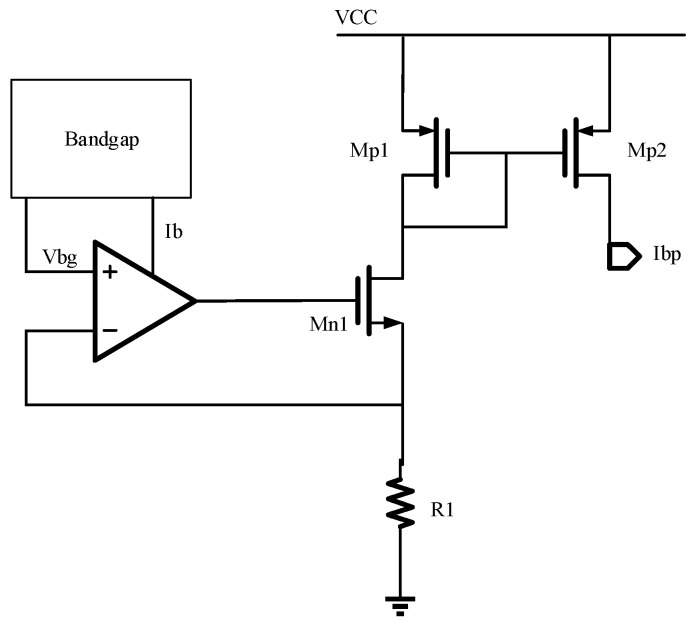
The topology of the reference current circuit.

**Figure 13 micromachines-14-00561-f013:**
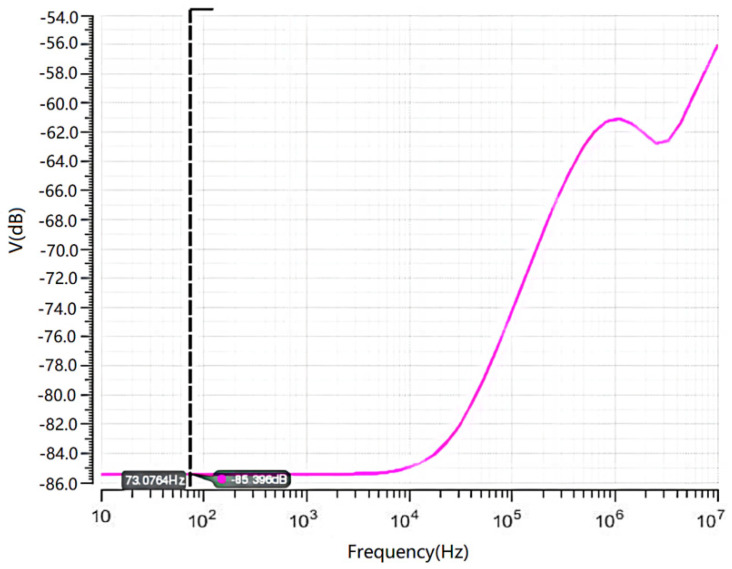
The PSR simulation diagram of the reference current circuit.

**Figure 14 micromachines-14-00561-f014:**
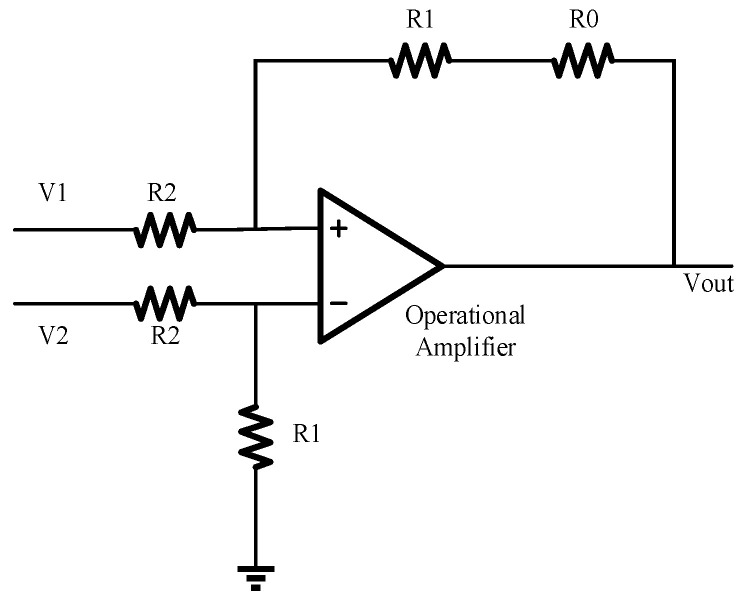
The improvement of the output offset.

**Figure 15 micromachines-14-00561-f015:**
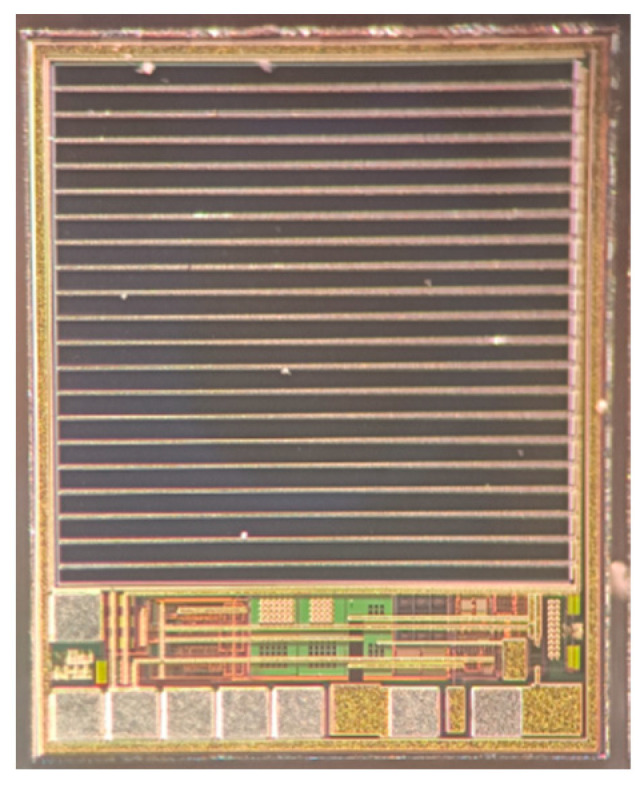
The micrograph of the chip.

**Figure 16 micromachines-14-00561-f016:**
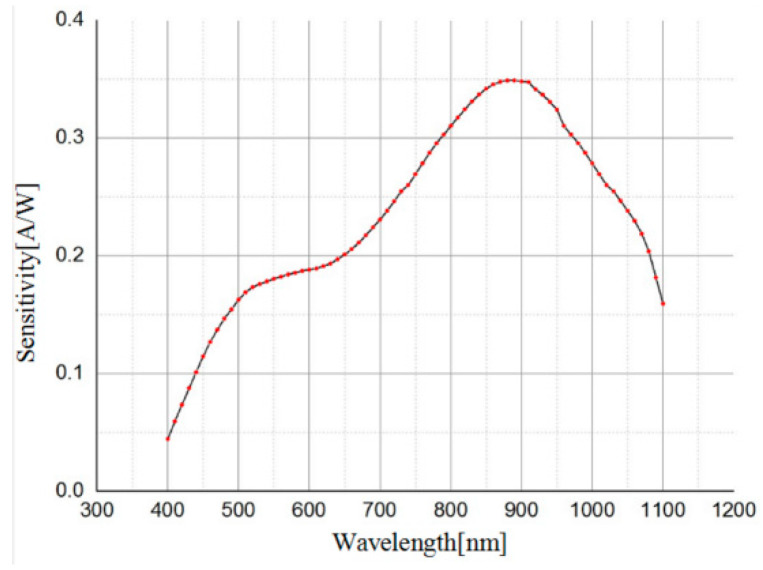
Measured spectral response of the PD.

**Figure 17 micromachines-14-00561-f017:**
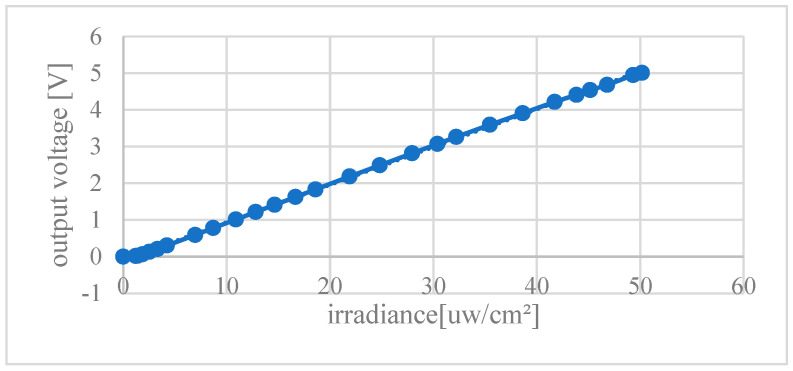
The curve of the output voltage versus LED irradiance.

**Figure 18 micromachines-14-00561-f018:**
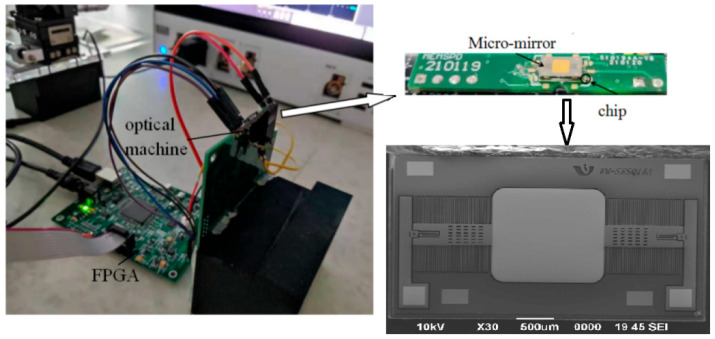
A photo of the experimental setup.

**Figure 19 micromachines-14-00561-f019:**
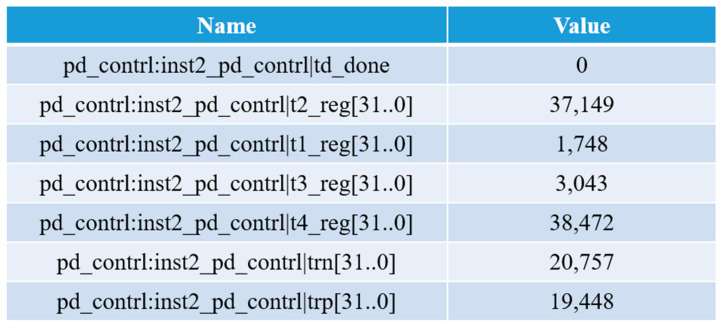
The signal captured by an FPGA logic analyzer.

**Figure 20 micromachines-14-00561-f020:**
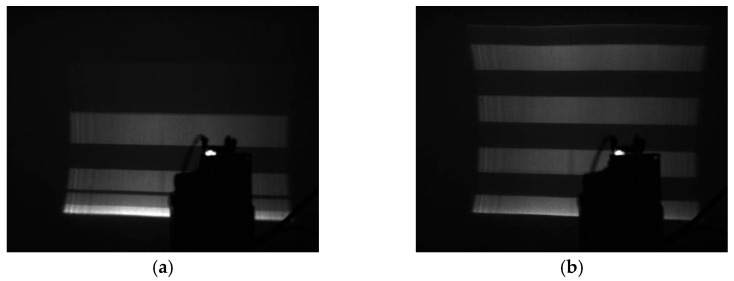
Projection binary fringes 10101010.

## Data Availability

Not applicable.

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
