# Peer review of "Development of an Optoelectronic Integrated Sensor for a MEMS Mirror-Based Active Structured Light System"

_micromachines, 2023, doi:10.3390/mi14030561_

Round 1
Reviewer 1 Report
This paper states that a new optoelectronic integrated chip with large-area PDs for 850 nm structured light systems have been proposed. In Section 1, the authors describe existing methods and their challenges for detecting the phase of micromirrors. Section 2 describes the structure of structured light systems and the phase diagram of micromirror. Section 3 presents simulation and spectral response measurement results for photodiodes. Section 4 discusses receiver circuit and post-amplifier design. Section 5 presents the experimental setup and results.
In Sections 3 and 4, the authors focus their discussion on the design of the photodiode and receiving circuit. However, as the authors state in Sections 1 and 2, the main purpose of this research seems to be to detect and control the phase of micromirrors. Considered from this point of view, the experimental results and discussion presented in Section 5 are insufficient. The authors should discuss how much the phase of the micromirror is improved in the proposed structured light system based on the measurement results, not only the photographs like Fig. 20.
Also, some text and figures may appear incomprehensible to readers. For example, Figures 11 and 14 are clearly clippings of PC screens, which are very difficult to see and unfriendly to readers. Also, there is no description in Fig. 16, and the configuration and dimensions of the chip are unknown.
At present, it seems that additional experiments and restructuring of the contents are necessary, and unfortunately I have no choice but to reject it.
Reviewer 2 Report
The authors report an optoelectronic integrated sensor to correct the phase error of the micromirror. This sensor has a large-area photodetector and a receiving circuit. The proposed sensor was fabricated using a 180 nm CMOS process. Based on experimental results, this sensor could decrease the phase error of the micromirror to 2.5 %. However, the authors must improve their manuscript by considering the following issues:
1.-The introduction section is short. This section should add more detailed information on the advantages and limitations of diverse methods to monitor and compensate the phase shifts of scanning MEMS mirrors.
2.-The introduction should include the proposed MEMS sensor's main advantages or scientific contribution compared with other sensors used to correct the phase error of the micromirrors.
3.- What are the main limitations of the proposed MEMS sensor?
4.- The authors could incorporate discussions about the influence of the environmental parameters (e.g., temperature, moisture, pressure) on the sensor performance.
5.- The equations shown in the manuscript are images. However, the equations must be written using suitable software.
6.- The authors must describe all the parameters used in the equations.
7.- The description of the working principle of the structured light system must be improved.
8.- The size of the labels used in Figure 1 is small.
9.- The quality and resolution of the Figures 1, 2, 3, 6, 7, 11, 14, 16, and 17 must be significantly enhanced.
10.- The description of the model used in Figure 5 must include more information on the boundary conditions and properties of the materials.
11.- A sub-section on the description of the fabrication process of the MEMS sensor should be considered.
12.- The fifth section should add more experimental results on the performance of the proposed system.
13.-The discussions must be improved by considering more experimental results.
14.- The conclusion section can be enhanced based on the above comments.
Reviewer 3 Report
The authors have presented an optoelectronic integrated sensor for a MEMS mirror-based active structured light system to compensate the phase lag and improve the micromirror’s performance. The sensors are made by using .18um CMOS process, with commercial potential. Here are my suggestions:
1. Since authors mention in the Introduction session that the main reasons causing the phase errors in micromirror are environmental temperature, moisture, and pressure, the authors may want to discuss the performance of their proposed circuits in different temperature, moister or pressure ranges in the Discussion session.
2. Figure 4,5, 6, 13 and 19 seem directly from the Simulation’s screenshot. The authors are suggested to remake these figures from the raw data for better resolutions and clarity.
3. The mirror image in Figure 18 may be replaced by its SEM image to show the micromirrors.
4. Although technical details are included in the article, authors should explain the novelty of the circuit to compensate the phase error for the micromirror operation.
5. How much phase lag time this circuit can compensate? Can authors discuss the robustness of this circuit in compensating the phase lag?
6. The authors are also encouraged to include a few related references from MDPI’s “micromachines”
Reviewer 4 Report
The paper presents an optimized photodetector manufactured with a CMOS process with integrated readout that can be used to compensate the phase error in MEMS micromirrors. The paper is generally well written and clear, and a lot of details are provided about the photodetector and its integrated readout circuit. The authors also present some tests proving the effectiveness of the device in compensating the phase error in a micromirror prototype and concluding that it works better than off-the-shelf devices that can be used for the same purpose. The level of innovation of the paper is moderate, since the problem and the method used to solve it (a photodetector) seem to be consolidated in the literature. Moreover, the integration of the CMOS and the MEMS chip is not described at all, although it seems arguable that they are fabricated with different processes. In which way is it done? The only information that the authors provide about it are contained in Fig. 2 in which it is clear that the micromirror is realized on a different chip than the photodetector. It would be interesting for the reader to know how such integration was carried out (flip chip technology or manual mounting). Also the technological relevance of the paper would be different depending of the integration type. Another comment relates to some typos that are still present in the paper: for instance, at line 75 “SILVOCA” should be “SILVACO” and also at lines 110-111 there is a sentence with some confusion between buried oxide and handle layer that should be corrected.
Round 2
Reviewer 2 Report
This manuscript version was improved by considering the reviewer's comments.
Author Response
Thanks.
Reviewer 3 Report
all the comments are addressed. the paper has been improved and can be considered for publication.